# A Service Evaluation of Migrants’ Experiences of Accessing Healthcare in an Infectious Diseases Clinic in Ireland

**DOI:** 10.3390/ijerph22101522

**Published:** 2025-10-04

**Authors:** Fergal Howley, Cassandra Barrett, Eoghan de Barra, Samuel McConkey, Cora McNally, Peter Coakley

**Affiliations:** 1Department of Infectious Diseases, Beaumont Hospital, D09 V2NO Dublin, Ireland; howleyfe@tcd.ie (F.H.);; 2Department of Social Work, Beaumont Hospital, D09 V2NO Dublin, Ireland; 3Royal College of Surgeons of Ireland, University of Medicine and Health Sciences, 123 St Stephen’s Green, D02 YN77 Dublin, Ireland

**Keywords:** migrant health, infectious diseases, refugee, direct provision

## Abstract

The healthcare needs of refugees and people seeking asylum are often broad and complex, with a higher burden of communicable diseases. There are limited data describing migrants’ experiences of accessing healthcare in Ireland. This cross-sectional study describes the experiences of migrants accessing healthcare services through an Irish Infectious Diseases clinic. Individuals attending the infectious diseases services in our hospital who had migrated to Ireland were included. Data were collected via a questionnaire, focusing on factors that may limit access to care, including communication, accessibility, cost, and stigmatisation. Seventy-six patients participated in this study. N = 20 (26%) of patients reported a commuting time of more than two hours to attend our clinic. N = 11 (15%) had experienced being unable to access healthcare in Ireland due to cost. Trust in healthcare providers was high (88%), and patient-reported satisfaction with communication was high (>90%). Persons living in direct provision services were more likely to report issues around privacy and less likely to have registered with a general practitioner. Accessibility and privacy were among the biggest challenges faced by migrants attending infectious diseases services at our centre, while communication and trust in healthcare providers were identified as areas of strength. Considering the burden of infectious diseases in migrant populations, and the challenges that certain migrant populations face in accessing healthcare, it is important to identify potential barriers to accessing care in order to ensure equitable, effective care. This study seeks to identify and describe the challenges that migrants face when accessing care through an Irish infectious diseases clinic. The results can help inform service provision and allocation of resources at a local level, while also identifying an area for further research regarding the barriers to accessing care faced by migrant communities in Ireland.

## 1. Introduction

The healthcare needs of migrants, and in particular of refugees and people seeking asylum, are often broad and complex. The burden of communicable diseases such as human immunodeficiency virus (HIV), viral hepatitis and tuberculosis (TB) is often higher in this cohort than in the general population of host countries [1]. Migrants also have increased mortality from communicable diseases, highlighting the need for increased access to infectious diseases services in migrant sub-populations [2]. Not only can this improve patient outcomes through reduced mortality rates, but it can also have public health benefits in reducing disease burden and transmission. Refugees and people seeking asylum have a particularly high rate of communicable diseases. In England, for example, an estimated 72% of new cases of TB are identified in refugees and people seeking asylum. This may be aggravated by prolonged periods in transit or time spent in poor living conditions, resulting in reduced access to healthcare and increased exposure to infectious diseases [1]. In addition, data from the United States of America (US) have identified refugees and migrant populations as having lower rates of immunisation and increased risk of vaccine-preventable disease, further compounding the risk of morbidity and mortality from communicable diseases in this patient cohort [3].

Patterns of healthcare utilisation among migrant populations vary widely accordingly to host country and migrant sub-groups. Irish data have identified lower rates of primary-care and tertiary-care utilisation among migrants (excluding those born in the United Kingdom) in Ireland, with difficulties navigating the complex healthcare service and concerns around cost cited as potential contributary factors. This is in keeping with data from the US [4].

In the twelve months from April 2022–April 2023, 141,600 immigrants arrived in Ireland, representing a 16-year high. Among these, nearly a quarter were returning Irish nationals, while almost 42,000 were Ukrainian [5].

Direct Provision is a system of state-funded accommodation for people seeking asylum in Ireland [6]. These centres, often re-purposed hotels or hostels, are dispersed across the country, and movement of individuals between centres is common. Centres are often crowded, with shared rooms and toilets, and lacking in self-catering facilities. Direct Provision services have been linked to poor physical and mental health, and inadequate provision of services. They have even been described by some reports as a violation of basic human rights. Irish research has also identified a lack of formal supports for provision of both social and primary-care services for those living in Direct Provision [6].

The need to provide ‘specific and comprehensive healthcare attention’ for refugees and asylum seekers in particular has been highlighted by the Royal College of Physicians of Ireland [7]. This need applies across myriad specialties, from primary care to maternity, medical, and mental health services. Despite this, studies have shown worse self-rated health among refugees when compared with the overall health of the Irish population [8], while others have described deficiencies in provision of maternity services for people seeking asylum in Ireland [9], and a high level of psychopathology among refugees in Ireland [10].

While there has been an increase in research focusing on migrant health in the Republic of Ireland in recent years, the majority of these studies focus on social determinants of health, public health preparedness, and health system adaptations [11]. There is less research on communicable diseases. Furthermore, there are limited data describing migrants’ experiences of accessing healthcare in Ireland.

We conducted this cross-sectional service evaluation to explore whether migrant patients attending our clinic face similar challenges to those described in the literature in accessing care.

The primary objective was to describe the experiences of migrants accessing healthcare services through the infectious diseases clinic in an Irish university teaching hospital, with an aim of identifying barriers to treatment, challenges faced, and areas for improvement in our current healthcare model of care provision for migrants.

## 2. Materials and Methods

### 2.1. Study Setting and Participants

Patients attending the dedicated TB, HIV and viral hepatitis infectious diseases (ID) clinic in Beaumont hospital who were identified as having migrated to Ireland from another country were included in this study. We used the United Nations definition of an international migrant as ‘any person who changes his/her country of usual residence’ [12]. Migrant status was determined either from medical records or during an appointment.

Exclusion criteria included patients attending clinic for the first time, and individuals who had emigrated from Ireland previously and subsequently moved back to Ireland.

Data collection took place over a two-month period (May–June 2023). Participants were given a questionnaire to complete themselves (or with assistance of a translator where required and available). The questionnaire focused on a number of areas including patient demographics, data regarding attendance at the ID clinic in our hospital (accessibility, frequency of attendances, years of attendance), and self-reported experiences of accessing healthcare in Ireland and in our hospital (including experience of prejudice or stigma, availability of translators where required, standard of communication). This included a series of closed-ended questions (e.g., multiple-choice questions), dichotomous questions (‘Yes/No’ questions), and open-ended questions (allowing respondents to answer in their own words). Participants were given privacy to complete the questionnaire to reduce the potential for bias.

We aimed for a sample size of 75 patients based on what we considered to be achievable within the available timeframe and sufficient to give a wide spectrum of experiences.

Approval for this study was granted by the Beaumont Hospital Audit Committee (registration number CA2023/097).

We used recommendations from a Royal College of Physicians of Ireland (RCPI) position paper on Migrant Health as a standard against which the service evaluation was conducted. In particular, we focused on the recommendations that translation services, access to primary care and other services should be available, and that relocated individuals should be involved in the decisions that affect them. We also focused on factors that may limit access to care, including transport and cost [7].

### 2.2. Statistical Analysis

Descriptive analyses were conducted including frequencies and percentages for categorical data and means (standard deviation, SD) for continuous data. Statistical comparisons between those living or not living in direct provision and association with (i) the need to hide medications from those living with, (ii) inability to access healthcare in Ireland (without costs) and (iii) having a GP were conducted using Fisher’s exact test and chi-square statistics. Significance at *p* < 0.05 is assumed. Stata (v18) was used for analysis.

## 3. Results

### 3.1. Demographics

76 patients were included in this study after meeting eligibility criteria. 37 (48.7%) were female, and the mean age was 43.72 (SD 8.31) years. More than 60% described their migrant status on arrival in Ireland as either ‘refugee’ or ‘seeking asylum’. Cohort characteristics and migration status are reported further in Table 1.

### 3.2. Communication and Translation Services

14 patients (18%) listed English as their native language, with a further 27 native languages represented.

The majority of participants described themselves as being fluent in English (71%), with a further 22% describing themselves as being able to conduct a conversation in English. Only five patients (6.6%) described their fluency in English as ‘minimal’.

Seven patients (9.2%) reported that they had at one time required a translator when attending clinic but that a translator had not been available.

Despite this, very few patients identified problems with communication during appointments, with less than five percent of patients saying that they had ever left clinic with unanswered questions or without understanding their management plan, investigations, or underlying diagnosis (Table 2). The majority of patients (92%) said they knew how to contact the infectious diseases services if they had any queries or concerns related to their healthcare or appointments.

### 3.3. Accessibility and Utilisation of Infectious Diseases and Other Healthcare Services

N = 23 (30.2%) of patients described attending the infectious diseases clinic on at least four occasions in the past twelve months, which is above average for HIV and viral hepatitis services in our hospital. Furthermore, N = 20 (26%) of patients reported that the commute to our facility took more than two hours. Of those with a commute time of more than two hours, 40% had attended clinic four or more times in the past year. The majority of patients reported that they usually attend clinic alone (88%).

N = 11 (14.5%) had experienced being unable to access healthcare in Ireland due to cost. When given the opportunity to leave comments, five patients highlighted cost of travel or distance of travel as areas in which the services could be improved.

A further fourteen patients said they do not have a general practitioner (GP) in Ireland (19.2% of those who answered).

### 3.4. Trust in Healthcare Providers and Stigmatisation

Two patients (2.7%) reported that they had at least once felt stigmatised or judged by a healthcare provider in our hospital, while a further four patients (5.3%) said they had previously felt unable to discuss an issue related to their healthcare due to privacy concerns.

When asked more generally if they had confidence and trusted in healthcare providers in Ireland, 88% said they had confidence in healthcare providers, with 2.7% saying they ‘sometimes’ had confidence, and 9.3% saying they did not have confidence in healthcare providers in Ireland.

Approximately one third of patients reported living alone, with the rest living with either family (44%), friends (13%), partners (6.5%) or strangers (2.5%). N = 15 (19.7%) reported living in Direct Provision accommodation. Differences between those living in Direct Provision accommodation and those not are outlined in Table 3. Those living in Direct Provision were more likely to report a lack of privacy regarding their medication (*p* = 0.039), and less likely to have a GP in Ireland (*p* = 0.022).

### 3.5. Comments and Suggestions

When asked to provide feedback on suggested improvements that could be made to enhance patient experience when attending clinic, 51 participants provided feedback. Of these, 23 left positive comments regarding the services. The majority of these highlighted the clear communication, kindness, and friendliness of healthcare providers. ‘The experience is good from the day I started coming here, I never felt unattended’, ‘team are really amazing’, and ‘best place I’ve ever been, first class’ were among the positive comments. 17 participants highlighted waiting times as being an area in which services could be improved, while a further 5 participants reported issues around cost of travel and commuting distance to clinic. 6 participants requested greater privacy around appointments, such as using personalised numbers instead of patient names when calling patients from the waiting area, and avoiding sending clinic appointment reminders with the words ‘Infectious Diseases Clinic’ included on the letter.

## 4. Discussion

To our knowledge, this service evaluation is the first to describe the experience of migrant patients attending infectious diseases services in Ireland.

Among the migrant population attending our clinic, we report a high number of individuals who arrived in Ireland as refugees or people seeking asylum, many of whom have subsequently gained citizenship or long-term residency status.

We identify a number of challenges faced by migrants in accessing and engaging with health services. This included cost of accessing healthcare, lack of access to primary care services, long commuting times to attend clinic, and challenges around privacy among those living in direct provision.

The issue of affordability of health services for migrants in Europe is well documented, and infectious diseases have specifically been identified as having a disproportionate burden on certain groups of migrants [13].

Almost one in five participants did not have a GP, despite our clinic recommending that every patient should have a primary healthcare provider. The lack of access to a GP often results in physicians in our clinic being asked to investigate and manage chronic conditions, often unrelated to the infectious issue, without appropriate pathways for follow-up. By comparison, a study of refugees, migrants and asylum seekers’ experience of accessing primary healthcare in the United Kingdom found that just five percent were not registered with a GP [14].

Challenges such as transportation, waiting times, childcare, and financial hardship have been identified previously as barriers for refugees in accessing healthcare [15], and many of these are reflected in our findings. Long commuting times and waiting times were frequently highlighted as challenges by participants in our study, while financial barriers to accessing healthcare were also in evidence. Long waiting times can have implications for patient care in the form of missed appointments, poor adherence with medications, and patient dissatisfaction [16], which may result in inefficient treatment of infections.

Among the participants living in direct provision, we identify challenges that are particularly common in this cohort. This included a lack of privacy (with many patients feeling the need to ‘hide’ their medications from those they were living with), while people living in direct provision were also less likely to have a GP. Logistical challenges around accessibility and availability, and a perception of discrimination relating to immigration status have been highlighted as barriers to accessing primary care services among refugees [17], and the strain caused by the opening of Direct Provision centres, both on the residents living in these centres and the healthcare providers in local primary care services, has been documented in Ireland [6].

In spite of the challenges highlighted, a number of positive aspects surrounding patient experiences were identified. These included a high rate of confidence in healthcare providers in Ireland, high levels of continuity of care, and good communication during consultations with patients reporting a good understanding of their medical condition and management. Communication, continuity of care and confidence are considered the three main challenges in healthcare delivery to migrants in high-income countries [18], so these findings were encouraging.

Furthermore, a number of participants provided positive feedback regarding our clinic services in the ‘suggestions’ section of the questionnaire. Medical, nursing, and administrative staff were described as being ‘excellent’, ‘kind’, ‘welcoming’, and ‘amazing’, despite issues such as long waiting times being highlighted as an area for improvement.

Limitations of this study include the small sample size and the fact that, as a service evaluation in a single centre, it does not reflect migrants’ experiences at a regional or national level. We used a broad definition of the term ‘migrant’ that included anyone born outside of Ireland, resulting in a heterogenous cohort of patients from different countries, backgrounds, ethnicities and cultures, who had moved to Ireland for many different reasons. We recognise that by assimilating this diverse group as one patient cohort, this study does not identify the specific experiences of subgroups from different ethnicities, regions, or cultures.

As this was an opt-in questionnaire, there is likely to be bias, as those with lower levels of English fluency may have been less inclined to participate in an opt-in questionnaire, particularly if a translator was not available. Furthermore, the use of questionnaires may result in desirability bias, with participants giving answers they consider desirable or favourable.

Finally, as this questionnaire was conducted on attendance at clinic, patients who had disengaged from care or who had previously had negative experiences in attending health services may have been under-represented.

Ultimately, a larger, multi-site study would be required to overcome some of these limitations in order to more accurately describe the challenges migrant populations face in accessing infectious diseases services. This would help to identify particularly vulnerable sub-populations, as well as geographical areas that require expansion of services.

Allowing for these limitations, this study does highlight a number of obstacles to accessing healthcare among migrant patients attending our clinic, in particular, issues around waiting times and accessibility. There are a number of steps to consider at a local level to improve accessibility, such as adjusting clinic times for individuals travelling long distances to attend, providing transport vouchers for those without the means to travel, and strategies to reduce waiting times. Broader measures to address these obstacles might include expansion of infectious diseases services at a regional or national level, establishing satellite clinics in regions that are difficult to access and poorly served by public transport, and offering accommodation that is close to hospitals with infectious diseases services for people seeking asylum who have chronic communicable diseases.

## 5. Conclusions

This service evaluation describes the experiences of migrants accessing infectious diseases services in an Irish hospital. Accessibility and privacy were among the biggest challenges faced by patients, while communication and trust in healthcare providers were identified as areas of strength. These findings can help inform decision-making regarding allocation of resources at a local level, while identifying areas for further research regarding the complex care needs of migrants, in particular those seeking asylum and living in direct provision, that might instruct policy at a national level.

## Figures and Tables

**Table 1 ijerph-22-01522-t001:** Demographics.

	Total Cohort (n = 76)
Age, years; mean (SD)	- 43.7 (8.3)
Sex, female; n (%)	- 37 (48.7)
Primary diagnosis	- HIV 59 (77.6%) - Viral Hepatitis 10 (13.2%) - Other ^&^ - Prefer not to say 5 (6.6%)
Region of origin	- Sub-Saharan Africa 39 (51.3%) - North Africa 12 (15.8%) - Europe 10 (13.2%) - Asia 7 (9.2%) - South America ^&^ - Other (3.9%) - Prefer not to say ^&^
Migrant status on arrival in Ireland	- Seeking asylum 36 (47.4%) - Economic migrant 19 (25%) - Refugee 10 (13.2%) - Other (e.g., student) 7 (9.2%) - Prefer not to say ^&^
Length of time living in Ireland	- Less than one year 10 (13.2%) - 1–5 years 23 (30.3%) - 5–10 years 5 (6.6%) - More than ten years 38 (50%)
Current visa status	- Citizenship/long-term residency granted 42 (55.3%) - Refugee status granted 13 (17.1%) - Awaiting interview/decision from international protection office 8 (10.5%) - Work visa ^&^ - Other/prefer not to say 10 (13.2%)
Years attending the infectious diseases clinic	- Less than one year 18 (23.7%) - 1–3 years 16 (21.1%) - 3–5 years 10 (13.2%) - More than five years 32 (42.1%)

^&^ Numbers < 5 were not reported.

**Table 2 ijerph-22-01522-t002:** Patient-Reported Levels of Communication.

Question Regarding Communication	Patients’ Answers
Have you ever left an appointment with unanswered questions?	Yes 3 (3.9%) No 72 (94.7%) Missing 1 (1.3%)
Have you ever left an appointment without understanding your treatment plan, investigations or diagnosis?	Yes 3 (3.9%) No 71 (93.4%) Missing 2 (2.6%)
Have you ever been prescribed medication without understanding why?	Yes 1 (1.3%) No 74 (97.4%) Missing 1 (1.3%)

**Table 3 ijerph-22-01522-t003:** Comparing the Experiences of Patients Living in Direct Provision with Others.

	Total (N = 76)	Living in Direct Provision (N = 15)	Not Living in Direct Provision (N = 61)	*p* Value *
Felt the need to hide medications from people they lived with [N (%)]	35/75 (46.7%) ^a^	10/14 (71.4%) ^a^	25/61 (40.1%)	0.039 **^$^**
Unable to access healthcare in Ireland due to cost	11/76 (14.5%)	3/15 (20%)	8/61 (13.1%)	0.405 ^&^
Has a GP in Ireland [N (%)]	59/73 (80.8%) ^b^	9/15 (60%)	50/58 (86%) ^b^	0.022 **^$^**

* significance at *p* < 0.05 is assumed; ^a^ one participant did not answer; ^$^ based on chi-square test for association; ^&^ based on Fisher’s exact test; ^b^ three participants did not answer.

## Data Availability

Data supporting results available upon request from corresponding author.

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
