# Peer review of "A Service Evaluation of Migrants’ Experiences of Accessing Healthcare in an Infectious Diseases Clinic in Ireland"

_ijerph, 2025, doi:10.3390/ijerph22101522_

Round 1

Reviewer 1 Report

Comments and Suggestions for Authors

Thank you for the opportunity to review this work—a novel study focused on describing migrant experiences in an infectious disease clinic in Ireland, with particular attention to the challenges migrants face and areas for improvement.

This manuscript fills a gap in the health services literature, providing its readers with an analysis of patient-driven/centered data that is critical for addressing barriers to access in infectious disease care. However, there are sections of the manuscript that could be strengthened.

Specifically, the introduction, which does not provide sufficient information about the communicable disease burden and impacts at the individual and population level. Please add more content to provide context for readers—make it clear why they should care about accessibility of communicability disease care for migrants. This would also strengthen your purpose statement.

Another area of concern is the methods. Overall, the study is methodologically sound, but revisions are needed, which will affect the results. I recommend several major revisions. Please see all my comments below.

------------------------------------------------------------------------------------------

Abstract: Could be strengthened with a public health impact statement at the end.

Lines 36-39: This section reads as paradoxical. It seems as if “inadequate provision of services” should be moved in front of the bit about human rights violations.

Lines 67-73: I would like to see more specifics on the questionnaire (e.g., Likert-type items? Open response?), not necessarily for reproducibility but to inform other studies.

Table 3: The expected frequency of some cells are below 5, making chi-square impractical for a comparison of those unable to access healthcare in Ireland due to cost. Please use and report Fisher’s Exact Test for this row. This will show the exact probability and a more accurate result. Also, please indicate the significance level of the results in this table (e.g., p < 0.5 denoted with an asterisk).

Lines 145-155: The comments and suggestions subsection of the results was a pleasant surprise but would be more impactful if it included anonymized quotes. Please consider making this addition if possible.

Lines 158-160: The second sentence of the discussion is misleading because of the over generalization. Please be more specific about the data being collected from a specific clinic.

Lines 203-209: Thank you for noting the limited generalizability of the study (it is an important study regardless!). You should also consider the potential for desirability bias due to the use of patients’ self-reported data.

Author Response

Dear Reviewer,

Thank you for your thorough and insightful critique of our study. Your feedback and advice is much appreciated, and we agree with the suggestions you have made to improve the overall message and the quality our work.

In response to each individual comment:

Comment 1: Specifically, the introduction, which does not provide sufficient information about the communicable disease burden and impacts at the individual and population level. Please add more content to provide context for readers—make it clear why they should care about accessibility of communicability disease care for migrants. This would also strengthen your purpose statement.

Response 1: We have added additional context with reference to two systematic reviews to outline the increased burden of infectious diseases in migrant populations. We have also outlined why this is important at an individual level (increased mortality from infectious diseases) and a population/public health level (lower vaccination rates, increased burden of disease, risk of onward transmission). We have referenced an Irish study outlining challenges migrants face in accessing care at a national level to highlight how this has been identified as a problem. We have also re-worded our purpose statement at the end of the introduction to make it stronger. 

Comment 2: Another area of concern is the methods. Overall, the study is methodologically sound, but revisions are needed, which will affect the results. I recommend several major revisions. Please see all my comments below.

Response 2: I hope by addressing your specific points of feedback below that we have addressed your concerns around the methods section. 

------------------------------------------------------------------------------------------

Comment 3: Abstract: Could be strengthened with a public health impact statement at the end.

Response 3: We have included a public health impact statement at the end of the abstract. 

Comment 4:Lines 36-39: This section reads as paradoxical. It seems as if “inadequate provision of services” should be moved in front of the bit about human rights violations.

Response 4: We have re-structured this sentence as suggested to make it read more clearly. We have also expanded the paragraph to give readers a more clear idea of how the Direct Provision system operates. 

Comment 5: Lines 67-73: I would like to see more specifics on the questionnaire (e.g., Likert-type items? Open response?), not necessarily for reproducibility but to inform other studies.

Response 5: I have included a line explaining the types of questions used, including open-ended, closed-ended, and dichotomous questions. We did not use Likert-type items as part of the questionnaire. 

Comment 6: Table 3: The expected frequency of some cells are below 5, making chi-square impractical for a comparison of those unable to access healthcare in Ireland due to cost. Please use and report Fisher’s Exact Test for this row. This will show the exact probability and a more accurate result. Also, please indicate the significance level of the results in this table (e.g., p < 0.5 denoted with an asterisk).

Response 6: Thank you for highlighting this. We have updated the table and made reference to the use of Fisher's exact test in our methods section. 

Comment 7: Lines 145-155: The comments and suggestions subsection of the results was a pleasant surprise but would be more impactful if it included anonymized quotes. Please consider making this addition if possible.

Response 7: We had not thought to include direct quotes but it is a really nice idea and we have included a few choice quotes as suggested, thank you. 

Comment 8: Lines 158-160: The second sentence of the discussion is misleading because of the over generalization. Please be more specific about the data being collected from a specific clinic.

Response 8: We have amended this sentence to make it clear that we are speaking specifically about migrants attending our clinic who first arrived in Ireland as refugees or people seeking asylum, rather than making a comment more generally about migrants arriving in Ireland. Thank you for pointing out the potentially misleading statement. 

Comment 9: Lines 203-209: Thank you for noting the limited generalizability of the study (it is an important study regardless!). You should also consider the potential for desirability bias due to the use of patients’ self-reported data.

Response 9: A valid point, thank you, and one which we had not considered. We have included a line highlighting the potential for desirability bias. 

In summary, we thank you for your time and expertise in reviewing our article. We acknowledge your suggestions and areas of concern and have sought to address them accordingly. Should you have any further feedback or suggested changes we would be grateful to hear them.

Yours sincerely,

Fergal Howley (on behalf of the authorship team)

Reviewer 2 Report

Comments and Suggestions for Authors

Dear Authors, 

The article's main objective is as follows: "This service evaluation seeks to describe the experiences of migrants accessing 54 healthcare services through the infectious diseases clinic in an Irish university teaching hospital, with the aim of identifying barriers to treatment, challenges faced, and areas for improvement in our provision of care for migrants." In hospital and even business terms, the study's results are well-planned and presented. Even, I find the limitations of the essay, as pointed out by the authors, to be clarly compelling:

“the small sample size and the fact that, as a service 203 evaluation in a single centre, it does not reflect migrants’ experiences at a regional or na-204 tional level. We used a broad definition of the term ‘migrant’ that included anyone born 205 outside of Ireland, resulting in a heterogenous cohort of patients from different countries, 206 backgrounds, ethnicities and cultures, who had moved to Ireland for many different rea-207 sons. We recognise that by assimilating this diverse group as one patient cohort, this study 208 does not identify the specific experiences of subgroups from different ethnicities, regions, 209 or cultures. 210 As this was an opt-in questionnaire, there is likely to be bias, as those with lower 211 levels of English fluency may have been less inclined to participate in an opt-in question-212 naire, particularly if a translator was not available. 213 Furthermore, as this questionnaire was conducted on attendance at clinic, patients 214 who had disengaged from care or who had previously had negative experiences in at-215 tending health services may have been under-represented.”

To ensure that the work does not remain merely a clear statistical study, and for the purposes to convert it in a scientific article, I believe it is necessary to precisely outline the objective and contribution in terms of their scientific value. Or to share or explore how to overcome the limitations of sample quality (social diversity is lost even despite the highly variable origin of the patients, the lack of precision regarding what a migrant is), and/or space (only one hospital, in one country), and/or time (only this study), any thesis in this respect with your same study could become a scientific issue. Where is the focus of the scientific or academic contribution, the thesis of the text, the hypothesis, its objectives, research questions, and a clear justification? It would seem to be the treatment models for migrant patients with communicable diseases, but does it?

The same text indicates that its contributions could be located within this population group, to develop a structured approach to identifying obstacles (barriers) and enablers (facilitators) in healthcare delivery to migrants with transmission diseases.

In conclusion, ¿what does implies something more than a particular and unique survey kind of Health Services Research (HSR), or it just it, what is the main thesis, the original improvement?

Author Response

Dear Reviewer,

Thank you kindly for giving your time and expertise to review our study. Your feedback and advice is much appreciated, and we have sought to address your suggestions and comments in order to improve the quality of our manuscript. 

We have provided a response to each individual comment below:

Comment 1: To ensure that the work does not remain merely a clear statistical study, and for the purposes to convert it in a scientific article, I believe it is necessary to precisely outline the objective and contribution in terms of their scientific value. Or to share or explore how to overcome the limitations of sample quality (social diversity is lost even despite the highly variable origin of the patients, the lack of precision regarding what a migrant is), and/or space (only one hospital, in one country), and/or time (only this study), any thesis in this respect with your same study could become a scientific issue.

Response 1: We have expanded our introduction to set the context of why accessibility of healthcare is of particular importance to migrant populations, and outlining the challenges they may face in accessing services. We have highlighted why this is important at both an individual level (increased mortality rates from infectious diseases) and at a public health level (lower vaccination rates, higher burden of disease, risk of onward transmission). We have also added to our 'limitations' section to highlight the need for further research in order to overcome some of these limitations. 

Comment 2: Where is the focus of the scientific or academic contribution, the thesis of the text, the hypothesis, its objectives, research questions, and a clear justification? It would seem to be the treatment models for migrant patients with communicable diseases, but does it?

Response 2: We have expanded the abstract to include a public health impact statement. We have also added text in order to strengthen the purpose statement in our introduction, outlining our objectives and justification for doing this study more clearly. 

Comment 3: The same text indicates that its contributions could be located within this population group, to develop a structured approach to identifying obstacles (barriers) and enablers (facilitators) in healthcare delivery to migrants with transmission diseases.

Response 3: We have added to our 'Discussion' section to highlight the potential contributions of this study. This includes a focus on how our findings might be used in future to guide further research and to improve pathways for provision of health services for migrant populations with communicable diseases. 

Comment 4: In conclusion, ¿what does implies something more than a particular and unique survey kind of Health Services Research (HSR), or it just it, what is the main thesis, the original improvement?

Response 4:  By expanding our introduction and including a public health impact statement, as well as making changes to the 'Discussion' section as outlined above, we have sought to clarify the main thesis of our article. 

In summary, we thank you for your time and expertise in reviewing our article. We acknowledge your suggestions and areas of concern and have sought to address them accordingly. Should you have any further feedback or suggested changes we would be grateful to hear them.

Yours sincerely,

Dr Fergal Howley (on behalf of the authorship team)

Round 2

Reviewer 2 Report

Comments and Suggestions for Authors

Dear authors:

I believe the article has improved to some extent.

1. The scale has been better defined.

2. The references allow the results to be placed in a comparative context.

3. The intention to promote the health of a highly vulnerable migrant population is reaffirmed.